# Immunogenicity Analysis of the Recombinant *Plasmodium falciparum* Surface-Related Antigen in Mice

**DOI:** 10.3390/pathogens11050550

**Published:** 2022-05-07

**Authors:** Jia-Li Yu, Qing-Yang Liu, Bo Yang, Yi-Fan Sun, Ya-Ju Wang, Jian Jiang, Bo Wang, Yang Cheng, Qiu-Bo Wang

**Affiliations:** 1Department of Clinical Laboratory, Wuxi 9th Affiliated Hospital of Soochow University (Wuxi 9th People’s Hospital), Wuxi 214000, China; lqy19870310@hotmail.com; 2Laboratory of Pathogen Infection and Immunity, Department of Public Health and Preventive Medicine, Wuxi School of Medicine, Jiangnan University, Wuxi 214000, China; 6202805011@stu.jiangnan.edu.cn (J.-L.Y.); 6192805009@stu.jiangnan.edu.cn (B.Y.); 9862021032@jiangnan.edu.cn (Y.-F.S.); 3Wuxi Red Cross Blood Center, Wuxi 214000, China; 6202805013@stu.jiangnan.edu.cn (Y.-J.W.); jjyilou@hotmail.com (J.J.); 4Department of Clinical Laboratory, The First Affiliated Hospital of Anhui Medical University, Hefei 230000, China; wangbo@ahmu.edu.cn

**Keywords:** *Plasmodium falciparum*, surface-related antigen, immunogenicity, invasion inhibition

## Abstract

*Plasmodium falciparum*, mainly distributed in tropical and subtropical regions of the world, has received widespread attention owing to its severity. As a novel protein, *P. falciparum* surface-related antigen (PfSRA) has the structural and functional characteristics to be considered as a malaria vaccine candidate; however, limited information is available on its immunogenicity. Here, we expressed three fragments of recombinant PfSRA in an *Escherichia coli* system and further analyzed its immunogenicity. The results showed that rPfSRA-immunized mice produced specific antibodies with high endpoint titers (1:10,000 to 1:5,120,000) and affinity antibodies (i.e., rPfSRA-F1a (97.70%), rPfSRA-F2a (69.62%), and rPfSRA-F3a (91.87%)). In addition, the sera of immunized mice recognized both the native PfSRA and recombinant PfSRA, the rPfSRA antibodies inhibited the invasion of *P. falciparum* into the erythrocytes, and they were dose-dependent in vitro. This study confirmed PfSRA could be immunogenic, especially the F1a at the conserved region N-terminal and provided further support for it as a vaccine candidate against *P.*
*falciparum*.

## 1. Introduction

Malaria is a devastating disease caused by *Plasmodium*, which remains the most serious public health problems worldwide and accounts for approximately 627,000 deaths in 2020 [1]. Six *Plasmodium* species, namely *Plasmodium falciparum*, *Plasmodium vivax*, *Plasmodium malariae*, *Plasmodium ovale wallickeri*, *Plasmodium ovale curtisi*, and *Plasmodium knowlesi*, are known to infect humans [2], among which *P. falciparum* causes the most severe form of human malaria [3]. In recent years, the resistance of malaria parasites to antimalarial drugs has continued to increase, which has brought great challenges to the prevention and treatment of malaria [4,5]; in addition, the complexity of the *Plasmodium* life cycle and its genome sequence have hampered the development of malaria vaccines [6,7,8]. RTS,S is currently the only vaccine that has proved to be protective against clinical malaria infections during phase III clinical trials, but the protective effect diminishes over time [9,10]. Therefore, an effective malaria vaccine is still an urgent priority for preventing malaria.

The invasion of erythrocytes by *Plasmodium* is an attractive process to investigate for advance understanding of the basic biological characteristics, which involves multiple receptor–ligand interactions [11,12]. The binding of malaria parasite ligands to specific erythrocyte surface receptors is known to mediate a series of steps involving the initial attachment of merozoites to erythrocytes, followed by a reorientation and the formation of the junction [13]. One of the current strategies for protection against malaria is to reduce the fraction of the merozoite invasion, with the consequent reduction in the incidence of parasitemia and malaria [14]. *P. falciparum* invades erythrocytes through redundant independent pathways even in the absence of one or two ligand–receptor interactions [13]. Thus, a multicomponent and multistage vaccine approach may be required to achieve sufficient protection against malaria [15].

The *P. falciparum* encodes more than 5000 genes [16]; however, the function of minority-encoded proteins by these genes is well established [17]. For example, *P. falciparum* reticulocyte-binding protein homolog 5 (PfRH5), essential for erythrocyte invasion, is confirmed to have modest immunogenicity in natural *P. falciparum* infection [18]. As an important indicator of vaccine, more immunogenic *Plasmodium* proteins should be discovered. *P. falciparum* surface-related antigen (PfSRA; PlasmoDB ID: PF3D7_1431400), a novel protein, which is observed on the surface of merozoites and gametocytes, is predicted to possess a signal peptide and be exported to the surface of the parasite [19]. This protein possesses coiled-coil signatures, which can form a stable structure to trigger functional antibodies [20,21]. In addition, native PfSRA has been detected in multiple processed fragments through a subtilisin-like parasite protease called PfSUB-1 [22]; of these fragments, the processed 32-kDa fragment exhibits erythrocyte-binding activity [19]. Earlier studies have analyzed genetic diversity of the *pfsra* gene and found that in contrast to the high degree of conservation of the N-terminal region, the C-terminal showed polymorphisms because of selective pressure [23]. These findings indicate that PfSRA has the structural and functional characteristics to be a new vaccine target.

In this study, the immune response against PfSRA in mice and the inhibitory activity of anti-rPfSRA antibodies on *P. falciparum* invasion were measured. Considering the large molecular weight of PfSRA for protein expression, recombinant PfSRA-Fragment 1a, recombinant PfSRA-Fragment 2a, and recombinant PfSRA- Fragment 3a (rPfSRA-F1a, rPfSRA-F2a, rPfSRA-F3a) were constructed based on the conserved regions of PfSRA in different orthologues that possess coiled-coil signatures, covering the peptides from a previous study [19]. The fragments of rPfSRA-F1a and -F2a were contained within the 70 kDa protelytic fragment, and -F3a was in the 32 kDa protelytic fragment [19]. We showed that mouse anti-PfSRA IgG restricted parasite invasion in vitro and that PfSRA protein could be immunogenic. These findings provide further support for PfSRA as a vaccine candidate against *P. falciparum*.

## 2. Results

### 2.1. Characterization, Expression, and Purification of rPfSRA

According to the structural characteristics of *pfsra* shown in the PlasmoDB website, *pfsra* possesses 990 amino acids (aa), in which the first 24 aa are signal peptides, and the last 22 aa are glycosylphosphatidylinositol (GPI) anchors. PfSUB-1 cleavage sites were predicted in the study [19,24,25] (Figure 1A). The *pfsra*-F1a (aa 214–315, 303 bp), *pfsra*-F2a (aa 512–590, 234 bp), and *pfsra*-F3a (aa 789–893, 309 bp) fragments were amplified successfully from the full-length plasmid sequence of *pfsra* (Figure 1B). The three fragments were designed based on a previous study on synthesized peptides of PfSRA; in addition, the peptide antibodies have demonstrated their potential in growth inhibitory activity as well as in their ability to recognize native proteins [19].

The results from the SDS-PAGE and Western blot showed that the rPfSRA-F1a, rPfSRA-F2a, and rPfSRA-F3a proteins with His-tag were successfully expressed and migrated at approximately ~38, ~35, and ~38 kDa, respectively, under reducing conditions (Figure 2A,B). 

### 2.2. Mouse Sera Recognized Both the rPfSRA and Native PfSRA

The specificity of mice-derived antibody was confirmed by Western blot using the rPfSRA protein. The results indicated that the mice immunized with rPfSRA-F1a, rPfSRA-F2a, and rPfSRA-F3a could produce specific antibodies against the corresponding protein. Moreover, the sera from the PBS-immunized mice (negative control) did not recognize any recombinant protein (Figure 2C).

To explore whether the recombinant protein maintains its native activity, the sera from the immunized mice were used to identify the native PfSRA in the crude protein of *P. falciparum*. As expected, all antibodies from the immunized mice (rPfSRA-F1a, rPfSRA-F2a, rPfSRA-F3a) consistently recognized multiple processed fragments (17, 24, 38, 55 kDa) of the native protein from the blood stages, and anti-rPfSRA-F3a could detect the full-length of PfSRA (Figure 3A–C). Thus, the sera from the immunized mice contained native PfSRA antibody against the crude protein from the 3D7 strain.

### 2.3. rPfSRA-Induced Humoral Immune Response in Mice

The titers of specific IgG in the protein-immunized mice sera were detected by ELISA after the obtained mice sera were diluted in different proportions. Overall, rPfSRA-F1a, rPfSRA-F2a, and rPfSRA-F3a induced a high immune response in mice with endpoint titers ranging from 1:10,000 to 1:5,120,000 (Figure 4A).

PfSRA-specific IgG (rPfSRA-F1a, rPfSRA-F2a, and rPfSRA-F3a) were detected in the sera after two weeks of the initial immunization, and the immune response was assessed on days 14, 28, 35, and 49 postimmunization; the IgG levels continued to increase until day 49 after the initial immunization. No reactivity was observed in the sera from the PBS-immunized mice (Figure 4B). In addition, IgG antibodies were induced in all mice groups immunized with the recombinant protein. The average avidity index (AIs) of the anti-rPfSRA-F1a, anti-rPfSRA-F2a, and anti-rPfSRA-F3a IgG were 97.70%, 69.62%, and 91.87%, respectively (Figure 4C).

### 2.4. rPfSRA Did Not Play a Role in Cellular Immune Response

Only the lymphocytes of the rPfSRA-F3a-immunized mice apparently proliferated. In addition, a significant difference was observed between the rPfSRA-F3a-immunized mice and the positive control ConA group (*p* < 0.05). By contrast, the rPfSRA-F1a and rPfSRA-F2a groups had no significant proliferation of lymphocyte compared with the positive control group (*p* > 0.05) (Figure 5A). In addition, the results of flow cytometry showed that the levels of CD4^+^-IFN-γ and CD8^+^-IFN-γ in the immunized mice did not change significantly (*p* > 0.05) (Figure 5B,C).

### 2.5. PfSRA Antibodies Inhibited the Invasion of P. falciparum into the Erythrocyte In Vitro

The 3D7 strain of *P. falciparum* was used to evaluate the inhibitory effect of anti-PfSRA on the invasion of *Plasmodium* in vitro [26]. The results showed that anti-rPfSRA exhibited obvious inhibition at a dilution ratio of 1:10, and the inhibitory effect of immunized-mice sera was dose dependent (Appendix A). As a positive control, the heparin group showed obvious inhibitory effect compared with preimmune control (*p* < 0.001). Anti-rPfSRA-F1a (*p* < 0.001), anti-rPfSRA-F2a (*p* < 0.01), and anti-rPfSRA-F3a (*p* < 0.001) antibodies were significantly effective in inhibiting the invasion of *P. falciparum*, with inhibition rates of 31.14%, 24.28%, and 25.79%, respectively (Figure 6A,B).

## 3. Discussion

While the malaria parasite has multiple redundant pathways to mediate the invasion of erythrocyte, the biological functions of the molecular basis of the invasion remains elusive [13] and characterizing this mechanism is critical for malaria control. Through the proteolytic process, the protein is extensively modified to ensure that the merozoites successfully invade the erythrocytes [27,28,29]. The subcellular localization of PfSRA is not altered by proteolytic processing, and the fragments or the unprocessed forms enter the erythrocytes during invasion [19]. Given that the unstructured region of PfSRA is difficulty to express, three fragments of PfSRA were used to evaluate immunogenicity. In this study, we demonstrated that PfSRA showed immunogenicity; furthermore, anti-rPfSRA antibody inhibited the invasion of erythrocytes by *P. falciparum*, providing further evidence of future PfSRA vaccines.

IgG is essential for determining the quality of malaria immunity and inhibiting the growth of parasites [30,31]. To evaluate the immune protection of PfSRA, whether its recombinant complex retained the specificity of the native antigen was first investigated. The results showed that all the sera from the immunized mice could specifically recognize both the corresponding recombinant protein and the native PfSRA, indicating that a highly specific antigen was successfully constructed. Nevertheless, natively processed PfSRA in the recognition of the immunized mice sera differed from a prior study [19], except the full-length band. This condition may be attributed to fragmentation of the protein, or to a difference between the effect of the antibody obtained from rPfSRA and the synthetic peptide.

The ability of PfSRA to induce antibodies in mice was assessed using serially diluted sera, and the humoral immune response mediated by IgG was investigated. The specific IgG was detected in the mice on day 7 after the first immunization, and the IgG level continued to rise until day 49 postimmunization. In addition, the three fragments of PfSRA could induce high levels of specific antibodies, up to an endpoint titer of 1:5,120,000. Antibody affinity characterizes the ability of an antibody to bind with an antigen, which is an important parameter indicating the immune response [32]. Besides, an earlier study of *Plasmodium falciparum* merozoite surface protein 3 (PfMSP3) suggested that antibody affinity is related to the inhibition of parasite growth [33]. The present data revealed that the three fragments of PfSRA could induce high-affinity antibodies, especially the rPfSRA-F1a fragment, suggesting that the antibodies induced in mice bind tightly to PfSRA. Furthermore, as a protein expressed on the surface of merozoites, the high-affinity antibodies induced by PfSRA indicated their role in inhibiting the growth of *Plasmodium*. Protective antibodies and T cell-mediated immunity have been identified to be important in controlling the blood-stage infection [34]. However, the results demonstrated that there was no significant difference in lymphocyte proliferation between the groups, involving IFN-γ from CD4^+^ T cells and CD8^+^ T cells, indicating that PfSRA appeared to be dispensable for regulating cellular immune response in mice.

Invasion of the erythrocytes is a key step in malaria infection and an important target of a protective immune response. Blood-stage vaccines provide protection mainly by inducing high-titer functional antibodies against the target antigen of *Plasmodium* and mediate the protective effect by inhibiting the proliferation of parasites at the blood stage or the invasion into the erythrocytes [14]. Importantly, the functional antibodies play a more prominent role than affinity antibodies in mediating the invasion inhibitory effect. Here, we performed in vitro *P. falciparum* invasion assays and found that the anti-rPfSRA antibody significantly inhibited the *P. falciparum* invasion of a laboratory 3D7 strain with dose-dependence, especially the anti-rPfSRA-F1a antibody. However, the 32 kDa fragment located in the rPfSRA-F3a fragment exhibited erythrocyte-binding activity during *P. falciparum* invasion into the erythrocytes [19]. This phenomenon may be due to the fact that the peptides in the previous study could not stand for all erythrocyte-binding domains. The other part of the explanation might be that potentially, antibodies targeting the N-terminal could interfere with the binding of the 32 kDa fragment at the C-terminal to erythrocytes simply by sterical hindrance, as the previous study found that the antibodies of PfMSP1_83_ at the N-terminal blocked the processing-inhibitory activity of anti-PfMSP1_19_ located at the C-terminal [35].

Overall, given the importance of the humoral immune response, invasion inhibition and the conservation of the N-terminal, the present study was encouraging for the F1a fragment at the N-terminal to be a candidate molecule for malaria vaccines. 

There were limitations of this study, including the lack of patient plasma samples for further serological screening with immunized mouse sera to assess the antigenicity of PfSRA; in addition, it might be better to measure more cytokines related to cellular and humoral immunity. 

## 4. Materials and Methods

### 4.1. Construction of Pfsra Plasmid

The nucleotide sequence encoding the full-length of *pfsra* was obtained from the PlasmoDB website (PF3D7_1431400), synthesized by TianLin Biotech (Wuxi, China) with codon optimization for expression in *Escherichia coli* (*E. coli*) system, and cloned into the pET30a vector. The *pfsra*-F1a, *pfsra*-F2a, and *pfsra*-F3a fragments were amplified by polymerase chain reaction (PCR) from the full-length gene, and the primer sequences used are listed in Appendix A containing the homology arms and Flag-tag sequences.

PCR amplification was performed using a Mastercycler (Eppendorf, Hamburg, Germany) under the following program: denaturation at 98 °C for 3 min, followed by 35 cycles of 98 °C for 10 s, 55 °C for 30 s, and 72 °C for 1 min, and a final extension at 72 °C for 5 min. The PCR products were purified by 2% agarose gel electrophoresis and cloned into a pET32a vector. The constructed plasmids were verified by double digestion with *Bam*HⅠ (NEB, Ipswich, MA, USA) and *Xho*Ⅰ (NEB, Ipswich, MA, USA) restriction enzymes, and were sent to TianLin Biotech (Wuxi, China) for sequencing.

### 4.2. Expression and Purification of rPfSRA

Recombinant plasmids of *pfsra* were transformed into *E. coli* BL21 (DE3) pLysS cells (TransGen Biotech, Beijing, China) and then grown in Luria Bertani broth containing ampicillin (50 µg/mL) at 37 °C for 12 h. The culture was induced by 0.1 mM isopropyl *β*-d-1-thiogalactopyranoside (IPTG; TransGen Biotech, Beijing, China) when the optical density at 600 nm (OD_600_) reached 0.4–0.6, and was allowed to grow for another 8 h at 37 °C. Finally, the culture was harvested by centrifugation at 4000× *g* for 30 min, and the protein was purified by TianLin Biotech (Wuxi, China). 

### 4.3. Verification of the Purified Protein

The protein expression was verified by SDS-PAGE and the gel was stained with Coomassie brilliant blue (Beyotime, Shanghai, China) for visualization. The proteins were separated by SDS-PAGE and transferred onto polyvinylidene difluoride membranes (PVDF; Immobilon, Darmstadt, Germany). Membranes were blocked with 5% skim milk in Tris-buffered saline containing Tween-20 (TBST) for 2 h at room temperature, followed by incubation with HRP-conjugated mouse anti His-Tag mAb (1:5000 dilution; ABclonal, Wuhan, China) overnight at 4 °C. After washing the membrane thrice with TBST, enhanced chemiluminescence (ECL, NCM biotech, Suzhou, China) was used to visualize the bands. The chemiluminescent signal was detected by the gel imaging analysis system (Bio-Rad ChemiDoc MP, Hercules, CA, USA) and analyzed by Image J software (Bio-Rad ChemiDoc MP).

### 4.4. Immunization of Mice

Twenty 6-week-old female BALB/c mice (Cavens, Changzhou, China) were randomly divided into four groups (five mice per group). Mice in experimental groups were immunized with rPfSRA-F1a, rPfSRA-F2a, and rPfSRA-F3a, respectively. Then, 50 µg of rPfSRA in PBS was emulsified with complete Freund’s adjuvant (CFA; Sigma, San Francisco, CA, USA) at a volume ratio of 1:1 with a total volume of 200 µL in the prime boost. The mixture was intraperitoneally injected into the mice. Additionally, incomplete Freund’s adjuvant (IFA; Sigma, San Francisco, CA, USA) was administered on day 21 and 42 postimmunization to boost the immunization. Control mice were immunized with CFA/IFA in emulsification with PBS. Blood was collected from each mouse on day 0, 7, 14, 28, 35, and 49 postimmunization by bleeding the tail vein, and sera were isolated for antibody detection by Western blot. The sera were used as primary antibodies at a dilution of 1:1500 in PBS to identify rPfSRA, and then HRP-conjugated goat anti-mouse IgG (CWBio, Beijing, China) was used as secondary antibody.

### 4.5. Identification of Native PfSRA

The 3D7 strain of *P. falciparum* was preserved at the Laboratory of Pathogen Infection and Immunity (Jiangnan University, Wuxi, China). The parasites were grown at 37 °C in a mixed environment of 90% N_2_, 5% O_2_, and 5% CO_2_, and were maintained in RPMI Medium 1640 (Gibco, New York, USA) containing O^+^ human erythrocytes (4% hematocrit), HEPES (Meilunbio, Dalian, China), NaHCO_3_ (Meilunbio, Dalian, China), AlbuMax Ⅱ (Sigma, San Francisco, CA, USA), hypoxanthine (Sigma, San Francisco, CA, USA), and gentamicin (Solarbio, Beijing, China). 

The schizonts were purified by 60% percoll (Solarbio, Beijing, China) gradient centrifugation, and lysed in 0.1% saponin (diluted in PBS) for 5 min on ice with intermittent mixing. The lysed material was centrifuged at 15,000× *g* for 5 min and washed thrice with PBS. Then, the parasite lysate was collected and boiled in a SDS-PAGE sample loading buffer (Meilunbio, Dalian, China) for 7 min [13]. The total protein was separated on SDS-PAGE gel, and native PfSRA in the *P. falciparum* crude protein was captured by the anti-rPfSRA mice sera at 1:1000 dilution overnight and detected by HRP-conjugated goat anti-mouse IgG (CWBio, Beijing, China).

### 4.6. Determination of Antibody Specificity and Avidity

The antibody levels in the mouse serum were measured by enzyme-linked immunosorbent assay (ELISA). A total of 50 ng of rPfSRA in a coating buffer (15mM sodium carbonate and 35mM sodium bicarbonate) was coated on 96-well plates (Corning, NY, USA) overnight at 4 °C, and blocked with 5% skimmed milk in TBST for 2 h at room temperature. After washing thrice with 0.1% TBST, serially diluted sera (1:10,000–1:5,120,000) were added and the samples were incubated at room temperature for 2 h. HRP-conjugated goat anti-mouse IgG antibody (Southern Biotech, Birmingham, AL, USA) was diluted at 1:5000 and added into each reaction well to incubate for 1.5 h at room temperature. Then, 100 µL of 3,3′,5,5′-tetramethylbenzidine (Bey time, Beijing, China) was added into the wells as a substrate for color development. The reaction was stopped by adding 50 µL of 2 M H_2_SO_4_ and the absorbance was measured at 450 nm.

The determination of AI was performed similarly to the steps described above. The difference was that after the mice sera incubated, the 96-well plates in the experimental groups were incubated with TBST containing 6 M urea for 10 min, whereas the control group was incubated without urea. Urea played a role in separating the weak binding in the antigen–antibody complexes. According to the OD_450_, the AI was calculated as follows:AI (%) = (OD_450_ of a group with 6 M urea/OD_450_ of a group without 6 M urea) × 100

### 4.7. Lymphocyte Proliferation Assay

After the mice were sacrificed by cervical dislocation, the spleen was harvested and ground on a filter bag containing RPMI 1640 (Meilunbio, Dalian, China). The ground solution was filtered through a filter membrane into a 15 mL centrifuge tube and centrifuged for 5 min at 1500× *g*. The supernatant was discarded, and the cells were lysed with an erythrocyte lysis buffer (Beyotime, Beijing, China). PBS was added to terminate the reaction of erythrocyte lysis and lymphocytes were collected by centrifugation.

The lymphocytes from the mice immunized with rPfSRA and PBS were grown on 96-well plates (5 × 10^5^ cells/well) and were treated with 10 µL rPfSRA-F1a (5 µg/mL), 10 µL rPfSRA-F2a (5 µg/mL), 10 µL rPfSRA-F3a (5 µg/mL), or 10 µL ConcanavalinA (ConA; 2 µg/mL). The plates were incubated at 37 °C with 5% CO_2_ for 72 h. Then, 10 µL of Cell Counting Kit-8 (CCK8; Yeasen, Shanghai, China) was added to each well and incubated at 37 °C with 5% CO_2_ for 2 h, and the absorbance value was measured at 450 nm. 

### 4.8. Measuring the Proportion of IFN-γ-Positive Lymphocytes

The splenocytes from the immunized mice were grown on 12-well plates. Phorbol 12-myristate 13-acetate (PMA; Sigma, San Francisco, CA, USA), ionomycin (Solarbio, Beijing, China), and brefeldin A (Solarbio, Beijing, China) were mixed to stimulate the cell at 37 °C with 5% CO_2_ for 6 h. After centrifugation, the cells were washed with PBS, and stained with CD4-488 (1:200; BioLegend, San Diego, CA, USA) and CD8-APC (1:300; BioLegend) for 1 h in the dark. Then, 100 µL of fixative (BioLegend) was added to fix the cells for 20 min at 4 °C in the dark. A membrane washing buffer (BioLegend) was added to wash and resuspend the cells. Cells were incubated overnight at 4 °C with IFN-γ-PE (1:200; BioLegend) diluted in a transmembrance washing solution. The analysis of the proportion of IFN-γ among gated CD4^+^ and CD8^+^ T cells was performed by flow cytometry (BD, Franklin Lakes, NJ, USA).

### 4.9. Invasion Inhibition Assay In Vitro

An invasion inhibition assay was conducted as previously described [26]. The culture of *P. falciparum* was synchronized with 5% sorbitol (Meilunbio, Dalian, China) to obtain a highly synchronized ring stage culture. When the malaria parasites were in the late trophozoites or schizonts, the parasites were diluted to 1.5% parasitemia and cultured with 2.5% hematocrit in a 96-well plate (100 µL parasite culture) in the gassed incubation chamber. The sera from immunized mice obtained previously were heated in a water bath at 56 °C for 30 min to inactivate the complement and then was added to the reaction system at a serial dilution ratio. When newly invaded ring-stage parasites were found, the cells were washed with PBS and fixed with 0.025% glutaraldehyde (Aladdin, Shanghai, China). The parasites were stained with SYBR Green I (Invitrogen, Waltham, MA, USA) in PBS for 30 min at 37 °C in the dark. Erythrocytes were washed with PBS three times and resuspended in 400 µL PBS. Flow cytometry was used to analyze the infected erythrocytes, and at least 100,000 cells were analyzed per sample. The experiment was repeated three times for each sample. The preimmune serum was used as a negative control and heparin severed as a positive control. Invasion (Inv) inhibition rate was calculated as follows [26]:Inhibition rate (%) = (1 − Inv (experimental group)/Inv (positive group)) × 100

### 4.10. Statistical Analysis

The data were analyzed by GraphPad Prism 5.0 and Microsoft Excel 2016. An unpaired Student’s *t*-test was used to determine the significance of the differences, and *p* < 0.05 indicated statistical significance.

## 5. Conclusions

The rPfSRA-F1a located at the conserved region N-terminal showed high immunogenicity and effectively inhibited the invasion of *Plasmodium* into the erythrocyte. Thus, the F1a fragment may be developed as a vaccine against malaria. 

## Figures and Tables

**Figure 1 pathogens-11-00550-f001:**
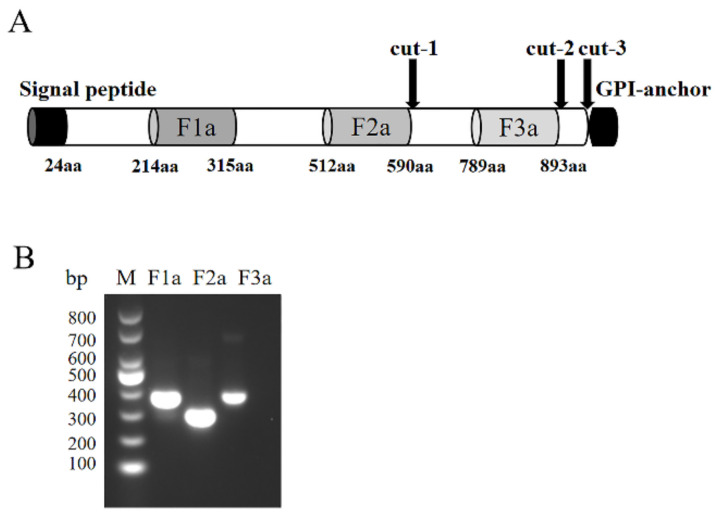
Schematic diagram and PCR amplification of PfSRA fragments. (**A**) Schematic diagram of PfSRA. The PfSRA protein contains 990 aa with a predicted signal peptide (1–24 aa) and a GPI anchor (969–990 aa). The PfSUB-1 cleavage sites are noted with black arrows. PfSRA-F1a fragment (214–315 aa), PfSRA-F2a fragment (512–590 aa), and PfSRA-F3a fragment (789–893 aa) were constructed for expression. (**B**) Fragmented *pfsra* amplification results.

**Figure 2 pathogens-11-00550-f002:**
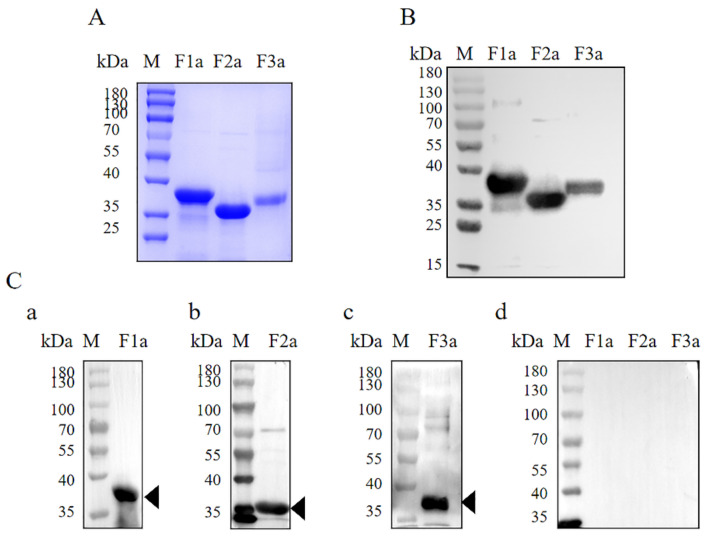
Expression and verification of PfSRA proteins. (**A**) The recombinant protein purification by SDS-PAGE was verified. rPfSRA-F1a (~38 kDa), rPfSRA-F2a (~35 kDa), and rPfSRA-F3a (~38 kDa). (**B**) An anti-His antibody was used to verify the protein expression by Western blot. rPfSRA-F1a (~38 kDa), rPfSRA-F2a (~35 kDa), and rPfSRA-F3a (~38 kDa). (**C**) The specificity of the immunized mice sera was detected by purified protein. The antibodies in sera from rPfSRA-immunized mice detected recombinant proteins, respectively, rPfSRA-F1a (a), rPfSRA-F2a (b), rPfSRA-F3a (c). The sera from PBS-immunized mice could not detect any fragment of rPfSRA (d).

**Figure 3 pathogens-11-00550-f003:**
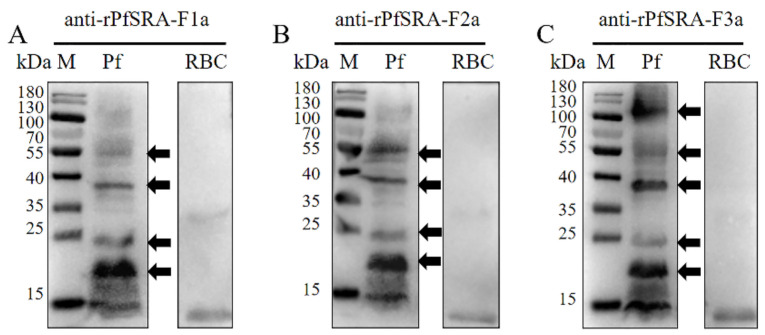
Expression of native PfSRA in *P. falciparum*. The multiple processed fragments of native PfSRA parasite protein were probed with rPfSRA-F1a mice antibodies (17, 24, 38, 55 kDa) (**A**), rPfSRA-F2a mice antibodies (17, 24, 38, 55 kDa) (**B**), and rPfSRA-F3a mice antibodies (17, 24, 38, 55, 113 kDa) (**C**). Uninfected erythrocytes were used as control.

**Figure 4 pathogens-11-00550-f004:**
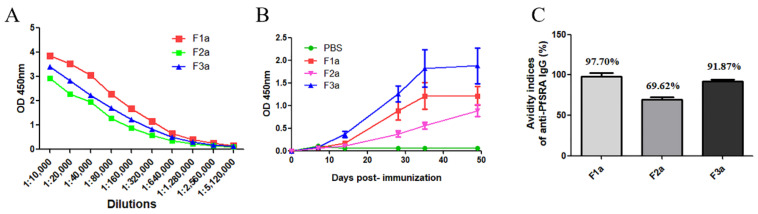
Humoral immune response in mice. (**A**) IgG antibody titer of anti-rPfSRA serum by ELISA. The x-axis shows the dilutions (dilution ranged from 1:10,000 to 1:5,120,000). (**B**) IgG levels in the rPfSRA-immunized mice were detected on day 7 postimmunization and the levels continued to rise. The sera of the PBS-immunized mice were set as the negative control. (**C**) Avidity index of anti-rPfSRA IgG antibodies to rPfSRA.

**Figure 5 pathogens-11-00550-f005:**
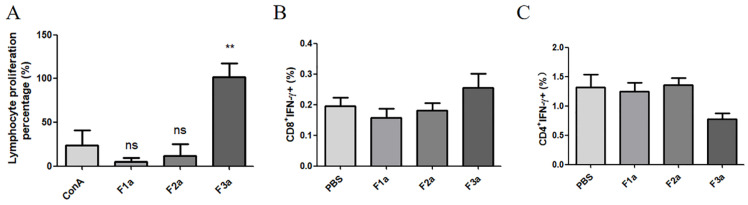
(**A**) Lymphocyte proliferation. Significant differences were found between ConA and F3a (** *p* < 0.01). However, no significant difference was found between ConA and F1a (*p* > 0.05), as well as ConA and F2a (*p* > 0.05). (**B**) and (**C**) showed the proportion of IFN-γ positive lymphocytes by flow cytometry. No significant changes were found in CD8^+^-IFN-γ and CD4^+^-IFN-γ (*p* > 0.05).

**Figure 6 pathogens-11-00550-f006:**
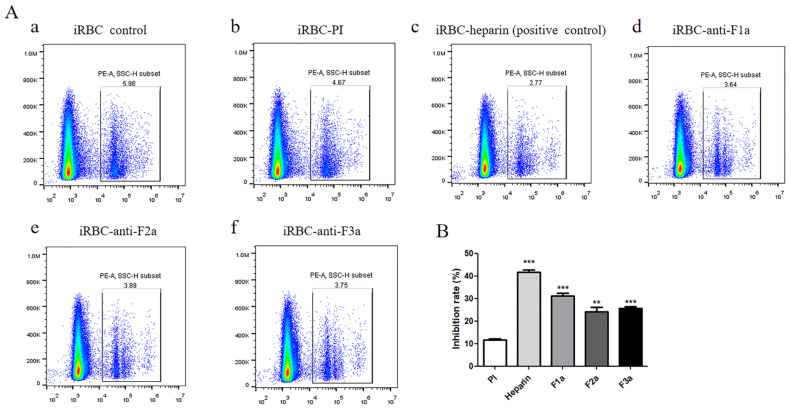
Anti-rPfSRA inhibits *P. falciparum* invasion into the erythrocyte. (**A**) The flow cytometry plots for one representative image from each treatment group are shown. Data above boxes represent the invasion inhibition rate. (**B**) Statistical analysis results. Inhibition rates of the pre-immune serum, positive control heparin, rPfSRA-F1a, rPfSRA-F2a, and rPfSRA-F3a groups were 11.63%, 41.73%, 31.14%, 24.28%, and 25.79%, respectively. All the inhibitory effects were statistically significant (*** *p* < 0.001, ** *p* < 0.01).

## Data Availability

Primer sequences used in PCR is contained in Appendix A. The primary data presented in this study are available on request from the corresponding author subject to applicable restrictions.

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
