# Peer review of "Immunogenicity Analysis of the Recombinant Plasmodium falciparum Surface-Related Antigen in Mice"

_pathogens, 2022, doi:10.3390/pathogens11050550_

Round 1

Reviewer 1 Report

Title: Immunogenicity analysis of the recombinant Plasmodium falciparum surface-related antigen in mice

Manuscript ID: pathogens-1631836

Review decision: 11 Mar, 2022

Reviewer comments:

The authors reported the analysis on the immunogenicity of the recombinant Plasmodium falciparum surface-related antigen in mice

However, the overall flow and the contents of this paper is not enough. In addition, the results are incorrect. Therefore, the authors have to clearly add or supplement or revise in the manuscript some points as follows:

1) Plasmodium falciparum is mainly distributed in Africa, but it is distributed in other area, such as the Middle East, Latin America, and Southeast Asia. Therefore, the authors have to mention it in the text.

2) The author has to correctly write full name of “RTS.S” in the main text (on page 1).

3) The author has to correctly write full name of “PfSUB-1”, “rPfSRA-F1a”, “rPfSRA-F2a”, and “rPfSRA-F3a” in the main text (on page 2). This omitting pattern is not good.

4) In Figure1 (B), the band results were shown that pfsra-F1a is 303 bp, pfsra-F2a is 234 bp, and pfsra-F3a is 309 bp. However, there are not the size markers between the band results (250 bp~500 bp). These results are not correct. Therefore, the authors have to make a new result using the detail marker including 300 bp. In addition, pfsra-F2a is 234 bp, but it is located above 250 bp size. This figure shows the incorrect result.

5) In Figure2 (A and B), these results are very strange. Because their result orders was divided respectively (on 4 page). I don’t understand this result pattern. In general, scientists indicate or place the result as follows through SDS-PAGE staining and Western blot respectively.

Marker, rPfSRA-F1a (38 kDa); rPfSRA-F2a (35 kDa); and rPfSRA-F3a (38 kDa).

Therefore, the authors have to show the results as one pattern, including three lines. This is not difficult. Do not divide them.

6) In Figure 2(A and C), (A) The sizes of the recombinant proteins (rPfSRA-F1a (~38 kDa), rPfSRA-F2a (~35 kDa) and 94 rPfSRA-F3a (~38 kDa)) are not correct in SDS-PAGE results.

In addition, (C) The sizes of the immunized mouse serum are not also correct in its location [F1a (~38 kDa); F2a (~35 kDa); and F3a (~38 96 kDa)]. The authors have to clearly confirm the results of the band size (on page 4).

7) In “2.2. Mouse sera recognized both the rPfSRA and native PfSRA”

The authors mentioned as follows: As expected, all antibodies from the immunized mice (rPfSRA-F1a, rPfSRA-108 F2a, rPfSRA-F3a) consistently recognized multiple processed fragments (17, 32, 46, and 58 kDa) 109 of the native protein from the blood stages, and anti-rPfSRA-F3a detected the full-length 110 of PfSRA (Figure 3A-3C).

However, in Figure 3 (A-C), the results are incorrect. I can’t find their band sizes (32, 46, and 58 kDa) in the figures. Why~? The authors have to indicate correctly the band sizes of proteins in figures (on page 5).

8) The author has to correctly write full name of “AIs” in the main text (line 132 on page 5).

9) The author has to correctly add the reference in the end of the following sentence; The 3D7 strain of P. falciparum was used to evaluate the inhibitory effect of anti-136 PfSRA on the invasion of Plasmodium in vitro (line 136~137 on page 5).

10) The author has to correctly write full name of “PfMSP3” in the main text (line 173 on page 6).

11) The author has to exactly write “5’- and 3’- ” in primers of the Supplementary Table S1. 

Author Response

Response letter

Dear Professor,

Thanks for your constructive criticisms provided on our manuscript. Herein we would like to submit revised manuscript as new submission entitled “Immunogenicity analysis of the recombinant Plasmodium falciparum surface-related antigen in mice” for publication in “Pathogens and upload a point-by-point response to your concerns. (Manuscript ID: pathogens-1631836)

I affirm that all authors listed in this manuscript have participated in and concur with the manuscript submission and subsequent revision.

Sincerely,

Yang Cheng

Laboratory of Pathogen Infection and Immunity, Department of Public Health and Preventive Medicine; Wuxi School of Medicine, Jiangnan University, Wuxi, Jiangsu, People's Republic of China

Phone No.: +8615190315127

Fax No.: +8651085328605

Title: Immunogenicity analysis of the recombinant Plasmodium falciparum surface-related antigen in mice

Response to reviewer 1 comments:

  • Plasmodium falciparumis mainly distributed in Africa, but it is distributed in other area, such as the Middle East, Latin America, and Southeast Asia. Therefore, the authors have to mention it in the text.

Response: Thanks for your suggestion, and we have added detailed area of Plasmodium falciparum distribution in the manuscript. (Line 22~23)

2) The author has to correctly write full name of “RTS.S” in the main text (on page 1). 

Response: Thank you for pointing this out. RTS,S, as a malaria vaccine, was named this way from the beginning. We are sorry for that we searched related articles and did not find the full name of RTS,S (PMID: 27355532; PMID: 25913272). (Line 49)

3) The author has to correctly write full name of “PfSUB-1”, “rPfSRA-F1a”, “rPfSRA-F2a”, and “rPfSRA-F3a” in the main text (on page 2). This omitting pattern is not good. 

Response: Thank you for pointing this out, and we have added the full name of them in the manuscript. (Line 77; Line 86~89)

4) In Figure1 (B), the band results were shown that pfsra-F1a is 303 bp, pfsra-F2a is 234 bp, and pfsra-F3a is 309 bp. However, there are not the size markers between the band results (250 bp~500 bp). These results are not correct. Therefore, the authors have to make a new result using the detail marker including 300 bp. In addition, pfsra-F2a is 234 bp, but it is located above 250 bp size. This figure shows the incorrect result.   

Response: Thanks for your careful review. The DNA markers used in our laboratory are all from TransGen Biotech in BeiJing (Trans2K plus DNA marker) with a distribution of 5000-3000-2000-1000-750- 500-250-100 bp. We are sorry that our laboratory does not have detailed DNA marker contain 300bp. The band in the figure is larger than the theoretical one, because we added the flag-tag (24 bp) when designing the primers, the size of band in the figure is larger than expected. Our PCR products have been sequenced and compared with the theoretical base sequence. The following are the blasting figures. If you think it is necessary, we will add them into the supplementary figures later. 

The first line of the three pictures is the theoretical base sequence, and the next few lines are the samples we sent for sequencing.

(1) F1a     

(2) F2a  

(3) F3a  

5) In Figure2 (A and B), these results are very strange. Because their result orders was divided respectively (on 4 page). I don’t understand this result pattern. In general, scientists indicate or place the result as follows through SDS-PAGE staining and Western blot respectively.

Marker, rPfSRA-F1a (38 kDa); rPfSRA-F2a (35 kDa); and rPfSRA-F3a (38 kDa).

Therefore, the authors have to show the results as one pattern, including three lines. This is not difficult. Do not divide them.

Response: Thanks for your suggestion. We have performed SDS-PAGE and Western blot, and put them together in one pattern. (Figure 2A, 2B)

6) In Figure 2(A and C), (A) The sizes of the recombinant proteins (rPfSRA-F1a (~38 kDa), rPfSRA-F2a (~35 kDa) and rPfSRA-F3a (~38 kDa)) are not correct in SDS-PAGE results.

In addition, (C) The sizes of the immunized mouse serum are not also correct in its location [F1a (~38 kDa); F2a (~35 kDa); and F3a (~38 kDa)]. The authors have to clearly confirm the results of the band size (on page 4).

Response: Thanks for your careful review. The results of Fig. 2A and 2B showed small differences for band size between F1a, F2a and F3a. In Figure 2C, the protein was seperated by 12% SDS-PAGE, which could separate more small size of protein molecular weight, so the ~35 kDa proteins were not separated obviously. Compared with protein marker, the band sizes were indeed between 35 kDa to 40 kDa, the molecular weights of these were approximately 38 kDa, 35 kDa, and 38 kDa.

7) In “2.2. Mouse sera recognized both the rPfSRA and native PfSRA”

The authors mentioned as follows: As expected, all antibodies from the immunized mice (rPfSRA-F1a, rPfSRA-108 F2a, rPfSRA-F3a) consistently recognized multiple processed fragments (17, 32, 46, and 58 kDa) of the native protein from the blood stages, and anti-rPfSRA-F3a detected the full-length of PfSRA (Figure 3A-3C).

However, in Figure 3 (A-C), the results are incorrect. I can’t find their band sizes (32, 46, and 58 kDa) in the figures. Why~? The authors have to indicate correctly the band sizes of proteins in figures (on page 5).

Response: Thanks for your careful review. The band sizes of proteins in figures were approximately 17, 24, 38, 55 kDa. According Amlabu’s study [19], native PfSRA is processed into multiple fragments (17, 32, 46, 58, 70 kDa) in parasite culture supernatant. However, the native PfSRA recognized by immunized-mice sera is not identical to this study. We think it may be due to the fragmentation of the protein when preparing the crude protein of Plasmodium falciparum, or it may be that the recombinant protein we used is longer than the synthetic peptide used in Amlabu’s study, and the antibody obtained from this recombinant protein recognizes the crude protein better.

8) The author has to correctly write full name of “AIs” in the main text (line 132 on page 5). 

Response: Thanks for your suggestion, and we have added the full name of AIs in the manuscript. (Line 134)

9) The author has to correctly add the reference in the end of the following sentence; The 3D7 strain of P. falciparum was used to evaluate the inhibitory effect of anti-PfSRA on the invasion of Plasmodium in vitro (line 136~137 on page 5). 

Response: Thanks for your suggestion, and we have added the reference in the end of the sentence. (Line 147)

10) The author has to correctly write full name of “PfMSP3” in the main text (line 173 on page 6).

Response: Thanks for your suggestion, and we have added the full name of PfMSP3 in the manuscript. (Line 184~185)

11) The author has to exactly write “5’- and 3’- ” in primers of the Supplementary Table S1.

Response: Thanks for your suggestion, and we have added 5’- and 3’- in primers of the Supplementary Table S1. in the manuscript. (Line 563)

Reviewer 2 Report

In this study, Yu  et al aimed at functional characterization a Plasmodium falciparum surface-related antigen. This is an interesting study but has multiple gaps that needs to be addressed to make this paper worthy publication.

  1. The author should give the details and expression profiles of other regions of PfSRA not just the three truncates described.

  1. Fig 2B: please show the uncropped figure 2B left panel (F1a). Were there some low molecular weight signal as seen in figure 2A?

  1. Figure 2C: uncropped images of figure 2C should be presented, at least as supplementary figures.

  1. Fig 2C: Instead of using sera from PBS immunized mice, the NEGATIVE control should be with sera pre-immunization for each mice since this sera is available (fig 4B).

  1. Line 128: no need of saying “strong” immune response.

  1. Line 132: All abbreviation should be written in full in their first appearance.

  1. What was the essence of lysing the schizonts, and then washing?

  1. How was the native PfSRA captured? In western blot?

  1. Since there were 5 mouse per group, which mouse was used this study? or did they use pooled sera?

  1. The authors need to include an additional figure showing the GIA activity of serially diluted purified mice anti PfSRA IgG with well-defined final IgG concentrations. It’s impossible to accurately compared the GIA run using sera of different animals.

  1. The authors claim that the N terminal is conserved, however, they don’t provide any evidence. Please provide figures and data related to this.

Author Response

Dear Professor,

Thanks for your constructive criticisms provided on our manuscript. Herein we would like to submit revised manuscript as new submission entitled “Immunogenicity analysis of the recombinant Plasmodium falciparum surface-related antigen in mice” for publication in “Pathogens and upload a point-by-point response to your concerns. (Manuscript ID: pathogens-1631836)

I affirm that all authors listed in this manuscript have participated in and concur with the manuscript submission and subsequent revision.

Sincerely,

Yang Cheng

Laboratory of Pathogen Infection and Immunity, Department of Public Health and Preventive Medicine; Wuxi School of Medicine, Jiangnan University, Wuxi, Jiangsu, People's Republic of China

Phone No.: +8615190315127

Fax No.: +8651085328605

Title: Immunogenicity analysis of the recombinant Plasmodium falciparum surface-related antigen in mice

Response to reviewer 2 comments:

  • The author should give the details and expression profiles of other regions of PfSRA not just the three truncates described.

Response: Thanks for your careful review. As a novel protein, PfSRA possesses coiled-coil signatures, signal peptide and GPI-anchor according to PlasmoDB website, which we have shown in introduction (Line 71~77) and Figure 1. More details of PfSRA has not clear and still need to be excavated.

  • Fig 2B: please show the uncropped figure 2B left panel (F1a). Were there some low molecular weight signal as seen in figure 2A?

Response: Thanks for your careful review, and the uncropped figure 2B was as follows. We have revised them together in one pattern.

 Original image of figure 2B            M  1  2  (see attached file)

M: marker (180-130-100-70-55-40-35 kDa); 1: rPfSRA-F1a; 2: other protein we expressed

  • Figure 2C: uncropped images of figure 2C should be presented, at least as supplementary figures.

Response: Thanks for your careful review. We detected the antibody specificity by western blot, and the immunized mice sera were used as the primary antibody to identify the corresponding recombinant protein, and PBS immunized mice sera as a control. Therefore, when incubating the serum, the membrane was cut and the corresponding serum was incubated according to different proteins. I’m sorry that there is no uncropped image. If you think it is necessary, we will re-perform this part of the experiment and add the figures later.

  • Fig 2C: Instead of using sera from PBS immunized mice, the NEGATIVE control should be with sera pre-immunization for each mice since this sera is available (fig 4B).

Response: Thanks for your careful review. We used PBS immunized mouse serum as a control, because PBS was added to each group and the volume adjusted for each protein volume to ensure a consistent volume ratio. In order to rule out the potential effect of PBS, we used PBS as a control. We refer to previous studies (PMID: 32562621) and think that using PBS as a control is the same as the pre-immune sera.

  • Line 128: no need of saying “strong” immune response.

Response: Thanks for your suggestion, and we have revised this sentence in the manuscript. (Line 129)

  • Line 132: All abbreviation should be written in full in their first appearance.

Response: Thanks for your careful review, and we have added the full name in the manuscript. (Line 134)

  • What was the essence of lysing the schizonts, and then washing?

Response: Thanks for your careful review. The essence of lysing with 0.1% saponin was to damage the Plasmodium falciparum-infected erythrocytes membranes and obtain the parasites, then washing and centrifugation was to remove the cell membrane debris.

  • How was the native PfSRA captured? In western blot?

Response: Thanks for your careful review. Yes, we performed western blot with the extracted crude protein of Plasmodium falciparum, and used the obtained serum of mice immunized with rPfSRA as the primary antibody to identify the native PfSRA in the crude protein.

  • Since there were 5 mouse per group, which mouse was used this study? or did they use pooled sera?

Response: Thanks for your careful review. Enzyme-Linked Immunosorbent Assay (ELISA), lymphocyte proliferation assay and measuring proportion of IFN-γ positive lymphocytes were performed from sera of 5 mice samples per group. Western blot and Invasion inhibition assay in vitro were performed from pooled sera of 5 mice per group.

  • The authors need to include an additional figure showing the GIA activity of serially diluted purified mice anti PfSRA IgG with well-defined final IgG concentrations. It’s impossible to accurately compared the GIA run using sera of different animals.

Response: Thanks for your suggestion. The IgG concentration was not known because the sera were not purified. The mice with the three recombinant proteins were immunized at the same time in the same controlled-environment, besides the reagents and protein doses were all the same. Therefore, we believe that it is feasible to conduct GIA comparison of the three groups. We also refer to other articles which directly use the unpurified serum of immunized animals for experiments. (PMID: 35202655) When selecting the serum ratio, we did make a series of concentration dilutions. And this part of the data has been supplemented in the supplementary figure1.

  • The authors claim that the N terminal is conserved, however, they don’t provide any evidence. Please provide figures and data related to this.

Response: Thanks for your careful review. This part of the results has been published, so we were not provide related figures and data to claim that the N-terminal is conserved, and we have cited this article in the manuscripts (PMID: 34421996). The reference was as follows:

Yang B., Liu H., Xu Q.W., Sun Y.F., Xu S., Zhang H., Tang J.X., Zhu G.D., Liu Y.B., Cao J., Cheng Y. Genetic Diversity Analysis of Surface-Related Antigen (SRA) in Plasmodium falciparum Imported From Africa to China. Frontiers in Genetics. 2021, 12, 688606.

Reviewer 3 Report

The present study entitled "Immunogenicity analysis of the recombinant Plasmodium falciparum surface-related antigen in mice" described the potential of PfSRA vaccine candidate against P. falciparum. The study is well designed and presented. However, the authors should address the following comments:

Abstract: This section should be improved as it should be compact and more informative. The authors included a bit more background that should be removed.

Introduction: Provide some information about the other immunogenic molecules of Plasmodium. Also provide references about the immunogenicity of surface related antigen.

Methods: Immunization of mice. Why did the authors choose intraperitoneal route for immunization , why not subcutaneous route?

Did the authors assess the safety of the vaccine formulation?

Authors analyzed total IgG. Why did they not analyze various IgG isotypes?

Results: The data of the lymphocyte proliferation should be included in the main manuscript, not as a supplementary material?

The authors should have analyzed the data of important cytokines, including IFN-g, IL-4 and IL-2 etc.

Discussion. Include the limitations of the study in this section.

Author Response

Dear Professor,

Thanks for your constructive criticisms provided on our manuscript. Herein we would like to submit revised manuscript as new submission entitled “Immunogenicity analysis of the recombinant Plasmodium falciparum surface-related antigen in mice” for publication in “Pathogens and upload a point-by-point response to your concerns. (Manuscript ID: pathogens-1631836)

I affirm that all authors listed in this manuscript have participated in and concur with the manuscript submission and subsequent revision.

Sincerely,

Yang Cheng

Laboratory of Pathogen Infection and Immunity, Department of Public Health and Preventive Medicine; Wuxi School of Medicine, Jiangnan University, Wuxi, Jiangsu, People's Republic of China

Phone No.: +8615190315127

Fax No.: +8651085328605

Title: Immunogenicity analysis of the recombinant Plasmodium falciparum surface-related antigen in mice

Response to reviewer 3 comments:

  1. Abstract: This section should be improved as it should be compact and more informative. The authors included a bit more background that should be removed.

Response: Thanks for your careful review. We have revised the abstract in the manuscript. (Line 22~35)

  1. Introduction: Provide some information about the other immunogenic molecules of Plasmodium. Also provide references about the immunogenicity of surface related antigen.

Response: Thanks for your careful review, and we have added the information about other immunogenic molecules of Plasmodium in the introduction (Line 66~70). At present, there are few articles introducing PfSRA, and the only two articles about the functional characterization [19] and gene diversity of pfsra [23] which we have cited in the manuscript.

  1. Methods: Immunization of mice. Why did the authors choose intraperitoneal route for immunization, why not subcutaneous route?

Response: Thanks for your careful review. We choose intraperitoneal route based on other articles for the use of intraperitoneal injection when immunizing mice (PMID: 33199351). In addition, considering that the absorption efficiency of intraperitoneal injection is faster than subcutaneous, and we attempt to induce an acute rather than chronic response.

  1. Did the authors assess the safety of the vaccine formulation?

Response: Thanks for your careful review. Yes, the protein was detoxified to reduce damage to the host before emulsification. Adjuvants may improve vaccine action, and elicit immune response.

  1. Authors analyzed total IgG. Why did they not analyze various IgG isotypes?

Response: Thanks for your suggestion. We also think it would be interesting to explore the IgG subclass and IgG1/IgG2a in immunized mice. But, we believe that the detection of total IgG can indicate the production of antibody against rPfSRA in immunized mice. Studies have shown that in mice, Th1 type produces IgG2a antibody and Th2 type produces IgG1 antibody. BALB/c mice usually produce Th2 immune responses to vaccine and subunit vaccine, which is related to the stimulation of IgG1 antibody (PMID: 32629975). Some studies have also shown that IgG1 antibody is much more than IgG2a antibody in immunized BALB/c mice (PMID: 30999945, PMID: 28532483). In the serum of patients infected with falciparum malaria, it was found that the highest production was IgG1 in the IgG subclass (PMID: 30930896). In the future research, we believe that it will be more meaningful to study the classification of IgG in the serum of patients infected with P. falciparum. Therefore, we have plan to explore the classification of IgG subclassed and IgG1/IgG2a in serum of patients infected with P. falciparum.

  1. Results: The data of the lymphocyte proliferation should be included in the main manuscript, not as a supplementary material?

Response: Thanks for your careful review. It was originally considered that PfSRA does not have effect on cellular immunity, and it was placed in the supplementary figure. We have now adjusted it into the main manuscript (Figure 5).

  1. The authors should have analyzed the data of important cytokines, including IFN-g, IL-4 and IL-2 etc.

Response: Thanks for your suggestion. We also think it would be interesting to analyze important cytokines including IL-4, IL-2 etc. IL-2 and IFN-γ are mainly produced by Th1 cells and mediate cellular immunity. IL-4 is mainly secreted and produced by Th2 cells to promote humoral immunity (PMID: 14517182). We mainly selected IFN-γ to evaluate cellular immunity, and IgG levels to evaluate humoral immunity. In future studies, we believe that it is more meaningful to improve the detection of cytokines related to cellular and humoral immunity.

  1. Include the limitations of the study in this section.

Response: Thanks for your careful review, and we have added the limitations of the study in discussion (Line 209~212).

Round 2

Reviewer 1 Report

Manuscript ID: pathogens-1631836 

Title: Immunogenicity analysis of the recombinant Plasmodium falciparum surface-related antigen in mice

The authors submitted revised manuscript as new submission entitled “Immunogenicity analysis of the recombinant Plasmodium falciparum surface-related antigen in mice” for publication in “Pathogens” and also upload a point-by-point response. 

However, the authors did not satisfy the basic results as follows:

(1) New comment as follows: 4) The author’s these responses and reason are not recognized scientific fields. If the authors have no the marker, buy the marker through the order. And then the authors have to confirm the results.

Previous comment: 4) In Figure1 (B), the band results were shown that pfsra-F1a is 303 bp, pfsra-F2a is 234 bp, and pfsra-F3a is 309 bp. However, there are not the size markers between the band results (250 bp~500 bp). These results are not correct. Therefore, the authors have to make a new result using the detail marker including 300 bp. In addition, pfsra-F2a is 234 bp, but it is located above 250 bp size. This figure shows the incorrect result.   

Response: Thanks for your careful review. The DNA markers used in our laboratory are all from TransGen Biotech in BeiJing (Trans2K plus DNA marker) with a distribution of 5000-3000-2000-1000-750- 500-250-100 bp. We are sorry that our laboratory does not have detailed DNA marker contain 300bp. The band in the figure is larger than the theoretical one, because we added the flag-tag (24 bp) when designing the primers, the size of band in the figure is larger than expected. Our PCR products have been sequenced and compared with the theoretical base sequence. The following are the blasting figures. If you think it is necessary, we will add them into the supplementary figures later. 

The first line of the three pictures is the theoretical base sequence, and the next few lines are the samples we sent for sequencing.

(1) F1a     

(2) F2a  

(3) F3a  

(2) New comment as follows: 5) The authors did not add the revised new results in the new submission manuscript.

Which place are the new results in the new manuscript~~?? I can not find the results.

Previous comment: 5) In Figure2 (A and B), these results are very strange. Because their result orders was divided respectively (on 4 page). I don’t understand this result pattern. In general, scientists indicate or place the result as follows through SDS-PAGE staining and Western blot respectively.

Marker, rPfSRA-F1a (38 kDa); rPfSRA-F2a (35 kDa); and rPfSRA-F3a (38 kDa).

Therefore, the authors have to show the results as one pattern, including three lines. This is not difficult. Do not divide them.

Response: Thanks for your suggestion. We have performed SDS-PAGE and Western blot, and put them together in one pattern. (Figure 2A, 2B)

(3) New comment as follows: 6) The author’s these responses and reason are not recognized scientific fields.

The authors have to buy the new detailed markers through the order as well as add the new results in the new version.

Previous comment: 6) In Figure 2(A and C), (A) The sizes of the recombinant proteins (rPfSRA-F1a (~38 kDa), rPfSRA-F2a (~35 kDa) and rPfSRA-F3a (~38 kDa)) are not correct in SDS-PAGE results.

In addition, (C) The sizes of the immunized mouse serum are not also correct in its location [F1a (~38 kDa); F2a (~35 kDa); and F3a (~38 kDa)]. The authors have to clearly confirm the results of the band size (on page 4).

Response: Thanks for your careful review. The results of Fig. 2A and 2B showed small differences for band size between F1a, F2a and F3a. In Figure 2C, the protein was seperated by 12% SDS-PAGE, which could separate more small size of protein molecular weight, so the ~35 kDa proteins were not separated obviously. Compared with protein marker, the band sizes were indeed between 35 kDa to 40 kDa, the molecular weights of these were approximately 38 kDa, 35 kDa, and 38 kDa.

(4) New comment as follows: 6) The author’s these responses and reason are not recognized scientific fields.

The authors have to add the new results in the new version.

Previous comment: 7) In “2.2. Mouse sera recognized both the rPfSRA and native PfSRA”

The authors mentioned as follows: As expected, all antibodies from the immunized mice (rPfSRA-F1a, rPfSRA-108 F2a, rPfSRA-F3a) consistently recognized multiple processed fragments (17, 32, 46, and 58 kDa) of the native protein from the blood stages, and anti-rPfSRA-F3a detected the full-length of PfSRA (Figure 3A-3C).

However, in Figure 3 (A-C), the results are incorrect. I can’t find their band sizes (32, 46, and 58 kDa) in the figures. Why~? The authors have to indicate correctly the band sizes of proteins in figures (on page 5).

Response: Thanks for your careful review. The band sizes of proteins in figures were approximately 17, 24, 38, 55 kDa. According Amlabu’s study [19], native PfSRA is processed into multiple fragments (17, 32, 46, 58, 70 kDa) in parasite culture supernatant. However, the native PfSRA recognized by immunized-mice sera is not identical to this study. We think it may be due to the fragmentation of the protein when preparing the crude protein of Plasmodium falciparum, or it may be that the recombinant protein we used is longer than the synthetic peptide used in Amlabu’s study, and the antibody obtained from this recombinant protein recognizes the crude protein better

Author Response

Response to Reviewer 1 comments:

(1) New comment as follows: 4) The author’s these responses and reason are not recognized scientific fields. If the authors have no the marker, buy the marker through the order. And then the authors have to confirm the results.

Previous comment: 4) In Figure1 (B), the band results were shown that pfsra-F1a is 303 bp, pfsra-F2a is 234 bp, and pfsra-F3a is 309 bp. However, there are not the size markers between the band results (250 bp~500 bp). These results are not correct. Therefore, the authors have to make a new result using the detail marker including 300 bp. In addition, pfsra-F2a is 234 bp, but it is located above 250 bp size. This figure shows the incorrect result.

Response: Thanks for your careful review, and we apologize for this problem. We have performed the PCR experiment by the detailed DNA marker including 300 bp and replaced the Figure1 B in the manuscript (Figure1 B). The bands in the figure were larger than theoretical, because we added the flag-tag when designing the primers. Detailed primer information was shown in Supplementary Table 1 (Line 546). 

(2) New comment as follows: 5) The authors did not add the revised new results in the new submission manuscript.

Which place are the new results in the new manuscript~~?? I can not find the results.

Previous comment: 5) In Figure2 (A and B), these results are very strange. Because their result orders was divided respectively (on 4 page). I don’t understand this result pattern. In general, scientists indicate or place the result as follows through SDS-PAGE staining and Western blot respectively.

Marker, rPfSRA-F1a (38 kDa); rPfSRA-F2a (35 kDa); and rPfSRA-F3a (38 kDa).

Therefore, the authors have to show the results as one pattern, including three lines. This is not difficult. Do not divide them.

Response: Thanks for your careful review, and we apologize for this problem. The previous figures were placed in a separate folder, not in the manuscript. We have added the figures in the new submission manuscript. (Figure 2A, 2B) (Line 496)

(3) New comment as follows: 6) The author’s these responses and reason are not recognized scientific fields.

The authors have to buy the new detailed markers through the order as well as add the new results in the new version.

Previous comment: 6) In Figure 2(A and C), (A) The sizes of the recombinant proteins (rPfSRA-F1a (~38 kDa), rPfSRA-F2a (~35 kDa) and rPfSRA-F3a (~38 kDa)) are not correct in SDS-PAGE results.

In addition, (C) The sizes of the immunized mouse serum are not also correct in its location [F1a (~38 kDa); F2a (~35 kDa); and F3a (~38 kDa)]. The authors have to clearly confirm the results of the band size (on page 4).

Response: Thanks for your careful review, and we apologize for this problem. We have checked the protein markers on the market, and there are no more detailed ones. We referred to related articles involving Western blot, the protein marker is similar to ours (PMID: 26823464; PMID: 25583518). The sizes of recombinant protein are about ~35 kDa and ~38 kDa. The protein markers we used have 35 kDa and 40 kDa markers, which can explain the size of rPfSRA. We re-experimented with 8% SDS-PAGE to better separate the recombinant proteins (Figure 2C) (Line 496). And the results could not be placed in one pattern because the recombinant proteins were identified by their immunized mice sera, respectively.

(4) New comment as follows: 6) The author’s these responses and reason are not recognized scientific fields.

The authors have to add the new results in the new version.

Previous comment: 7) In “2.2. Mouse sera recognized both the rPfSRA and native PfSRA”

The authors mentioned as follows: As expected, all antibodies from the immunized mice (rPfSRA-F1a, rPfSRA-F2a, rPfSRA-F3a) consistently recognized multiple processed fragments of the native protein from the blood stages, and anti-rPfSRA-F3a detected the full-length of PfSRA (Figure 3A-3C).

However, in Figure 3 (A-C), the results are incorrect. I can’t find their band sizes in the figures. Why~? The authors have to indicate correctly the band sizes of proteins in figures (on page 5).

Response: Thanks for your careful review, and we apologize for this oversight. In our study, immunized mice sera were used to identify native PfSRA, and the bands (17, 24, 38, 55, 113 kDa) we identified were not completely consistent with Amlabu’s study [19]. Previously, we repeated the experiment and obtained a similar result. We speculated that this phenomenon may be attributed to fragmentation of the protein when preparing the crude protein of Plasmodium falciparum, or it may be that the recombinant proteins we constructed were longer than the synthetic peptide in Amlabu’s study, and the antibody obtained from this recombinant protein recognizes the crude protein better. In Amlabu’s study, we found that there will be a certain difference in the band size when antibodies obtained from different synthetic peptides recognize native PfSRA (PMID: 29912472).

Reviewer 3 Report

The authors have responded to my comments. I am satisfied with their response.

Author Response

Thanks for your constructive criticisms provided on our manuscript. Herein we would like to submit revised manuscript as new submission entitled “Immunogenicity analysis of the recombinant Plasmodium falciparum surface-related antigen in mice” for publication in “Pathogens and upload a point-by-point response to your concerns. (Manuscript ID: pathogens-1631836)

I affirm that all authors listed in this manuscript have participated in and concur with the manuscript submission and subsequent revision.